# Spatial–Temporal Distribution and Ecological Risk Assessment of Microplastics in the Shiwuli River

Lei Hong [1,2,3,4], Xiangwu Meng [1], Teng Bao [1,2], Bin Liu [1], Qun Wang [1], Jie Jin [2] and Ke Wu [2,3,*]

1   School of Biology, Food and Environment, Hefei University, Hefei 230601, China; honglei@hfuu.edu.cn (L.H.)
2   Hefei Institute of Environmental Engineering, Hefei University, Hefei 230601, China
3   Collaborative Innovation Center for Environmental Pollution Control and Ecological Restoration of Anhui Province, Hefei 230601, China
4   Anhui Key Laboratory of Sewage Purification and Ecorestoration Materials, Hefei 230001, China
*   Correspondence: wuke@hfuu.edu.cn

**Abstract:** This study aimed to investigate the distribution of microplastics (MPs) within the Shiwuli River in Hefei, a Chinese inland city. Water and sediment samples were collected during flood season (from May to September) and non-flood season (from October to April) at 10 representative points along the truck stream. The electron microscope, the laser direct infrared chemical imaging system (LDIR), and the scanning electron microscope (SEM) were used to observe and quantify the colour and shape of the MPs, to identify the number, size, and polymer composition of the MPs, and to observe the microstructures of typical MP particles, respectively. The polymer risk index (RI) model and the pollution load index (PLI) model were used to assess the polymer-related risks and the overall extent of MP pollution in the river, respectively. Analysis of MP abundance for different sampling points showed that the water of Shiwuli River had an average abundance of MPs of 8.4 ± 2.5 particles/L during the flood season and 5.8 ± 1.7 particles/L during the non-flood season; the sediment had an average abundance of MPs of 78.9 ± 8.3 particles/kg during the flood season and 63.9 ± 7.1 particles/kg during the non-flood season. The abundance of MPs of different points was investigated. Result show that the more abundances of MPs were found at confluences with tributaries (S4, S5, and S6), where they are also close to the residential and industrial development, while lower values were found in agricultural areas (S8) and wetland ecological regions (S9 and S10). In water, the maximum appeared at S5 with 21.7 ± 4.6 particles/L during the flood season and 15.9 ± 4.2 particles/L during the non-flood season, respectively; the minimum appeared at S9 with 1.8 ± 1.0 particles/L during the flood season and 2.2 ± 0.4 particles/L during the non-flood season, respectively. In sediment, the maximum appeared at S5 with 174.1 ± 10.1 particles/kg during the flood season and 143.6 ± 10.4 particles/kg during the non-flood season, respectively; the minimum appeared at S8 with 10.3 ± 2.8 particles/kg during the flood season and at S9 with 12.1 ± 3.2 particles/kg during the non-flood season, respectively. MP characteristics were also studied. Results show that the MPs mainly exhibited a fibroid morphology (27.90–34%), and red-coloured particles (19.10%) within the smaller size less than 500 μm (38.60%) were more prevalent. Additionally, the result of LDIR scanning shows that a total of eleven types of MP polymers were found in the river water and sediment, including acrylates (ACR), chlorinated polyethylene (CPE), ethylene vinyl acetate (EVA), polyethylene (PE), polyethylene terephthalate (PET), polypropylene (PP), polystyrene (PS), polyurethane (PU), polyvinylchloride (PVC), polyamide (PA), and silicon. The most common particle was PE (19.3–21.6%). Furthermore, the environmental risk assessment demonstrated that the PS polymer posed a Level-III risk in the water samples and a Level-II risk in the sediment samples from the Shiwuli River. The remaining polymer types exhibited Level-I risk. The $PLI_{zone}$ value for water was 2.24 during the flood season, indicating heavy pollution, and 1.66 during the non-flood season, indicating moderate pollution. Similarly, the $PLI_{zone}$ value for sediments was 2.34 during the flood season and 1.91 during the non-flood season, both suggesting a heavy pollution. These findings highlight the potential risk posed by MP pollution in the Shiwuli River to the quality of drinking water sources in Chaohu Lake in Hefei. They provide valuable insights into management, pollution control, and integrated management strategies pertaining to MPs in urban inland rivers in Hefei.





**Keywords:** microplastics; spatial–temporal distribution; ecological risk assessment

## 1. Introduction

Microplastics (MPs), defined as plastic particles smaller than 5 mm in size, represent persistent organic pollutants and emerging contaminants [1,2] that exhibit resistance to degradation and are ubiquitously present in various environmental compartments, including surface water, sediment, atmosphere, and soil, thereby posing significant ecological risks [3]. In general, MPs can be derived from both industrially produced primary plastic particles and secondary plastic particles generated through the fragmentation of large plastic debris commonly used in daily production and consumption [4]. MPs possess distinctive surface characteristics and display a phenomenon known as the Trojan horse effect [5]. This effect enables them to act as carriers for the adsorption of heavy metals, antibiotics, and other pollutants in the environment, facilitating their migration. Moreover, MPs can be ingested by aquatic organisms and mammals [6–8]; subsequently causing bioaccumulation through the food chain and potentially impacting human health [9–11]. These issues became widely concerning around the world.

Recent research predominantly concentrated on elucidating MP transport mechanisms within marine environments [12–14], assessing their consequential toxicological impacts on various organisms [15,16], and investigating their distribution in freshwater lakes and rivers [17–19]. Drabinski et al. explored the abundance and spatial dispersion of MPs in three distinct freshwater rivers situated in Rio de Janeiro, Brazil [20]. Similarly, Pradit et al. examined the quantification of MP particles in the U-Taphao River located in southern Thailand, investigating temporal variations across different time intervals [21]. Furthermore, Lin et al. explored the abundance and distribution patterns of MPs in the Pearl River, China [22]. These studies collectively shed light on the natural migratory pathways of MPs originating from urban industrial processes and residential activities as they traverse riverine ecosystems.

Various methodologies and models were developed to assess the risk associated with MP pollution in aquatic ecosystems. For instance, Kim et al. employed a species sensitivity distribution (SSD) method to evaluate the ecological risk of MP pollution by determining the sensitivity of soil biota to microplastics [23]. Similarly, Zhong et al. utilized a pollution load index (PLI) model to determine the level of risk posed by MPs in Dongshan Bay, China [24]. Furthermore, Ranjani et al. applied the PLI model to evaluate the presence of MPs in sediments from the east and west coasts of India [25]. In a separate study, Kai et al. utilized a polymer risk index (RI) model to assess the extent of MP pollution in Chagan Lake and Xianghai Lake, China [26]. By employing these diverse models, researchers paved the way for conducting comprehensive ecological risk assessments pertaining to MP pollution in water basin environments. These assessments, in turn, provide valuable insight into effective preventative measures and controls against MP pollution.

The Shiwuli River, located within Hefei, serves as a primary tributary of Chaohu Lake. Chaohu Lake is a significant freshwater lake in China with a water area of 769.6 km$^2$ and is recognized as a national drinking water source. This river connects Chaohu Lake with the urban area of Hefei, the capital city of Anhui Province, with a population of 9,634,000. The drainage basin of the Shiwuli River encompasses diverse functional zones, including administrative and commercial areas, industrial and agricultural production areas, residential living areas, as well as ecological landscape and wetland ecological protection areas. The river faces substantial risks from both point source and non-point source pollution, which act as major pathways for the enrichment and migration of MPs and other pollutants into Chaohu Lake. Currently, there is limited research focusing on the characteristics and ecological risk assessment of MPs in urban rivers situated inland, especially those that flow into drinking water source lakes. Additionally, no published studies investigated the ecological risk assessment of MPs in Chaohu Lake and its tributaries. Herein, this

study aims to investigate the spatial and temporal distribution of MPs in both river water and sediment within the Shiwuli River. Furthermore, it aims to quantitatively assess the potential ecological risks and overall pollution load using two models: the pollution load index (PLI) and the polymer risk index (RI). The ultimate goal is to provide a scientific foundation for comprehensive pollution assessment and source control management of MPs in the Chaohu Lake basin.

## 2. Materials and Experimental Methods

### 2.1. Profile of the Sampling Points

The Shiwuli River Basin is located at 31.727°–31.835° N, 117.198°–117.377° E, and covers an area of 111.25 km²; (Figure 1). The population in the basin is currently approximately 576,500. The precipitation during the flood season accounts for 60.5% of the total precipitation. The main stream of the Shiwuli River is 24.74 km long, the average slope of the watercourse is only 0.72‰, and the water surface ratio is only 4.4%. At present, the water resource utilization pattern is watercourse ecological water. There is no stable clean water source at the upper reaches and water supplementation mainly comes from natural precipitation, tailwater discharged from sewage treatment plants, and overflows from Swan Lake. The water volume cannot meet the basic ecological flow requirements. At present, point source pollution in the river comes mainly from the discharges of urban sewage plants and the effluent from industrial plants. Non-point sources are mainly urban surface runoff, rural domestic wastewater, and agricultural planting and cultivation. Many detrimental environmental factors have an impact on the basin, including incomplete rain pollution distributaries, a sharp increase in non-point source pollution during the flood season, and watercourse cut-off during the dry season, among others [27].

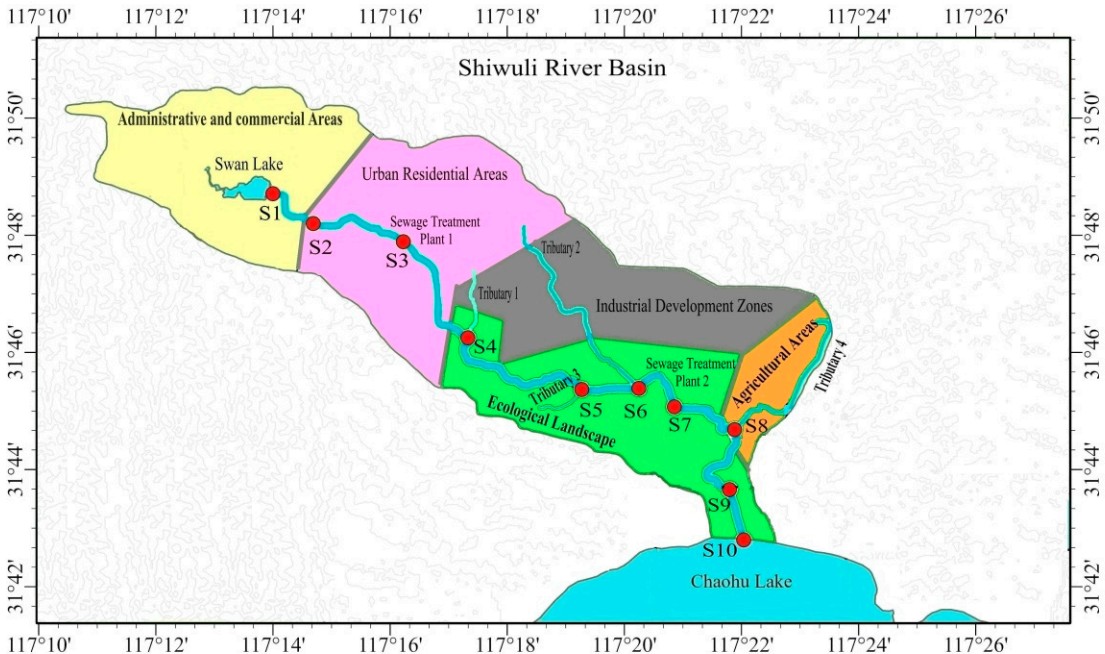

**Figure 1.** Sampling points and functional areas in Shiwuli River Basin. The sampling points selected along the river are labelled as S1–S10.

### 2.2. Sample Methods

According to spatial factors, such as the different urban functional areas that the Shiwuli River runs through, the type of outlets and the confluence of the tributaries, a total of 10 sampling points were chosen. Specifically, one sampling point was at the starting and one at the ending sections of the river, respectively, four points were at the intersection of the tributaries, two points were at outlet areas of sewage treatment plants, and three points

were at typical functional regions (Figure 1 and Table 1). Sampling was performed during the flood season (June 2022) and the non-flood season (November 2022). Samples of both water and sediment were obtained at each sampling point. A total of 2 L water samples were collected from each sampling point at three different depths (0–20 cm, 40–80 cm, and 100–150 cm) and uniformly mixed; 1.5 kg of sediments was collected at three different locations from each sampling point and uniformly mixed.

**Table 1.** Description of sampling points.

| Sampling Points | Locations | Characteristics |
| --- | --- | --- |
| S1 | Administrative and commercial areas | Starting section, estuary of Swan Lake |
| S2 | Urban residential areas | Beside main streets of urban traffic, outlet below the Overpass of Jinzhai Road |
| S3 | Urban residential areas | Outlet of the Hudaying Sewage Treatment Plant |
| S4 | Ecological landscape areas | Intersection with the tributary Xingfu Channel, next to the industrial development zone |
| S5 | Ecological landscape areas | Intersection with the tributary Wangniangou, the planned large ecological park area |
| S6 | Ecological landscape areas | Intersection with the tributary Xuxiaohe, the planned large ecological park area |
| S7 | Ecological landscape areas | Outlet of the Shiwulihe Sewage Treatment Plant |
| S8 | Agricultural areas | Intersection with the tributary Xuxi River, agricultural planting and aquaculture |
| S9 | Wetland ecological protection areas | Large water area |
| S10 | Wetland ecological protection areas | Terminal section, estuary of Chaohu Lake |

*2.3. Experimental Scheme*

2.3.1. Microplastics Separation and Extraction

The treatment of the samples was carried out according to the method of Meng et al. [28]. For water samples, stainless steel sieves with different pore sizes (1 mm, 5 mm, 15 μm, and 25 μm) were piled up one by one for coarse filtering. All the above sieves were rinsed by deionized water and filtered again using a filter membrane (LONGJIN, PTFE, aperture 5 μm, diam 50 mm, Nantong, China). Then, the filter membrane was placed in an open beaker containing 30% $H_2O_2$ solution for digestion and stirred by a magnetic stirrer (HUXI, HMS-203D, Shanghai, China) for 72 h. The temperature was 60–65 °C and the rotating speed was 550 rpm. The digested mixture was extracted for a second time and the wet filter membrane was placed in a petri dish, which was then transferred into an oven for dehydration for 1 h at 100–105 °C. The dried filter membrane was transferred into a saturated NaCl solution for density floating. Meanwhile, oscillation was carried out and the solution was left static for 24 h. The supernate was extracted and the residues in the beaker were poured into the saturated NaCl solution again for density floating. This process was repeated three times and three pieces of filter membrane were obtained. For the sediment samples, they were paved onto a petri dish and dried in an oven at 105 °C for 14 h. The subsequent digestion steps were the same as described above for the water samples.

2.3.2. Observation and Identification

The electron microscope (AOSVI, HK830–5870, China) was used to observe and quantify the colour and shape of the MPs; the laser direct infrared chemical imaging system (LDIR, Agilent 8700, Santa Clara, CA, USA) was used to scan all particles from the filter membranes and the Agilent Clarity software (version 1.1.2) in this system analyzed the MPs in each sample automatically and individually combined with its own spectral library (Microplastic starter 1.0) to obtain all polymer types with a matching degree greater than 0.80, as well as the number and size of particles in each type [29]. In addition, the microstructures of different types of typical MP particles were observed using a scanning

electron microscope (SEM, Hitachi S4800, Tokyo, Japan). All of the above were carried out in a closed and dust-free environment.

Quantification of the characteristics and components of the MPs was conducted with reference to the quantitative analysis method of Pivokonsky et al. [30].

$$N_m = \frac{\sum_{i=0}^{5} * N_i * S_m}{5 S_f} \tag{1}$$

$N_i$ is the number of MPs on each quadrate (particles/L, particles/kg), $S_m \approx 9.26$ cm$^2$ is the contact area of impurities on a single high reflector, and $S_f = 0.84$ cm$^2$ is the area of a single s quadrate. The length of a quadrate is 12.5 mm and its width is 6.72 mm.

### 2.4. Potential Ecological Risk Assessment

The polymer risk index (RI) model was used to assess the polymer-related risks of MPs pollution in the river. The ecological risk index $H$ of MP polymers was calculated as follows:

$$H = \sum P_n \times S_n \tag{2}$$

$P_n$ is the proportion of different MP types in the samples from each point, and $S_n$ is the risk scores of the different types of MP polymers [31].

The pollution load index (PLI) model was used to assess the overall extent of MP pollution in the river. The abundances of MPs at the regional sample points were used as the major indices according to the PLI model, which was proposed by Tolminson et al. [32]. The assessment model was defined as follows:

$$CF_i = C_i / C_{oi} \tag{3}$$

$CF_i$ was defined as the ratio of the abundance of MPs ($C_i$) at each sampling point to the minimum abundance of MPs ($C_{oi}$) at each sampling point.

$$PLI_i = \sqrt{CF_i} \tag{4}$$

$$PLI_{zone} = \sqrt[n]{PLI_1 \times PLI_2 \times \ldots \times PLI_n} \tag{5}$$

$PLI_i$ is the pollution load index of MPs for a single sample, while $PLI_{zone}$ represents the pollution load index of MPs for the river.

The risk level classification for the two models is shown in Table 2.

**Table 2.** Risk level classification for the two models [31,32].

| Model (Indexes) | Risk Category | | | |
| --- | --- | --- | --- | --- |
| | I (Very Low Hazard) | II (Low Hazard) | III (Medium Hazard) | IV (High Hazard) |
| RI ($H$ value) | <10 | 10~100 | 100~1000 | >1000 |
| PLI ($PLI_{zone}$ value) | <10 | 10~20 | 20~30 | >30 |

### 2.5. Data Analysis

Microsoft Excel 2019 was used for data pre-processing and SPSS 26.0 (IBM Co. Ltd., Armonk, NY, USA) for correlation analysis. Origin 2018 (Origin Lab., Farmington, ME, USA) and Microsoft Visio 2016 were used for data analysis and graph plotting.

## 3. Results and Discussion

*3.1. Spatial–Temporal Distribution of Microplastics*

### 3.1.1. Abundance Distribution

Figure 2 shows the result of MPs in water and sediment samples from the 10 points (S1–S10). The abundances of MPs in water and sediment were described by the number of MP particles/L and the number of MP particles/kg dry weight, respectively.

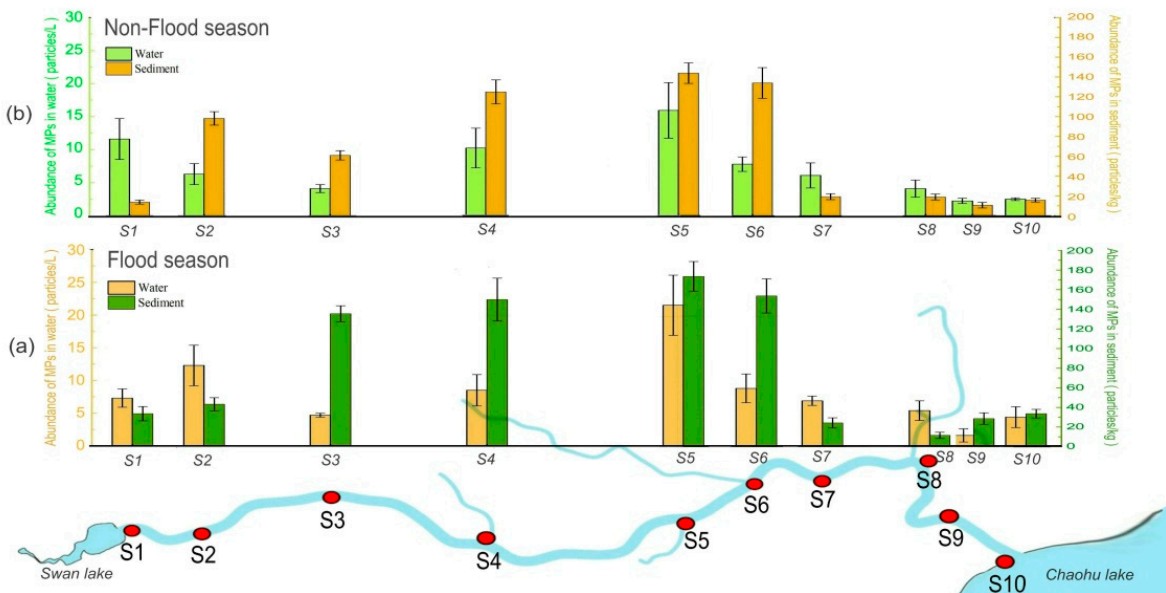

**Figure 2.** Distribution of microplastics abundance at sampling points along the Shiwuli River. The sampling points selected along the river were labelled as S1–S10. (**a**) Flood season; (**b**) Non-flood season.

In water, the average MPs abundance during the flood season was 8.4 ± 2.5 particles/L. The maximum was at S5 (21.7 ± 4.6 particles/L), followed by S2 (12.5 ± 3.1 particles/L), and the minimum was at S9 (1.8 ± 1.0 particles/L). The average MPs abundance during the non-flood season was 5.8 ± 1.7 particles/L. The maximum also was observed at S5 (15.9 ± 4.2 particles/L), followed by S1 (11.6 ± 3.1 particles/L), and the minimum also was at S9 (2.2 ± 0.4 particles/L). In sediments, the average MPs abundance was 78.9 ± 8.3 particles/kg during the flood season. The maximum was at S5 (174.1 ± 10.1 particles/kg) and the minimum was at S8 (10.3 ± 2.8 particles/kg). The average MPs abundance was 63.9 ± 7.1 particles/kg during the non-flood season. The maximum also was at S5 (143.6 ± 10.4 particles/kg) and the minimum was at S9 (12.1 ± 3.2 particles/kg).

When combining the statistical results for the water and sediment samples, it can be seen that the abundance of MPs was generally higher during the flood season (Figure 2a) compared to that during the non-flood season (Figure 2b).

The spatial variation laws of MP abundances in water and sediment were generally consistent and showed a trend of first increasing and then decreasing from upstream to downstream of the river [33]. The MPs were mainly concentrated in the middle and lower reaches of the river, influenced by the flow of the water [34]. The abundance of MPs in the sediment samples was higher than that in the water samples. This is mainly due to the fact that the water is poorly flowing in the middle reaches and stays stagnant for a long period of time [28]. MPs in the middle reaches can generate an absorption effect and can be wrapped by other substances, leading to their precipitation and accumulation over the years. Research showed that MPs can be easily wrapped in gravel [35]. When the water volume cannot meet the basic ecological flow requirement, MPs with a high density easily precipitate in water, while MPs with a low density may be adsorbed by algae, thus further increasing the ecological pollution load [14].

The abundances of MPs from S4, S5, and S6 were higher than those of the other points. Although these three points are located in the ecological landscape zone, they are all near residential living areas or industrial development areas. MPs produced by household waste and industrial waste are easily carried into rivers by surface rainfall runoff [18,36]. Sewage treatment approaches can intercept MPs to some extent. Therefore, the abundances of MPs in water at S3 and S7 were relatively low due to the influence of sewage treatment plant supplementation into the river. Chen et al. demonstrated that wetland environments have a significant effect on the removal efficiency of MPs. [37]. Due to the presence of wetlands, the water and mudflat areas at S9 are relatively large, providing better conditions for the natural degradation of MPs; this accounts for the low abundance of MPs at this point [38].

3.1.2. Shape Characteristics

According to previous studies, the shapes of MPs observed under the microscope were divided into the following categories: particles, membranes, fragments, and fibers. The shapes of the MPs observed via microscope are shown in Figure 3.

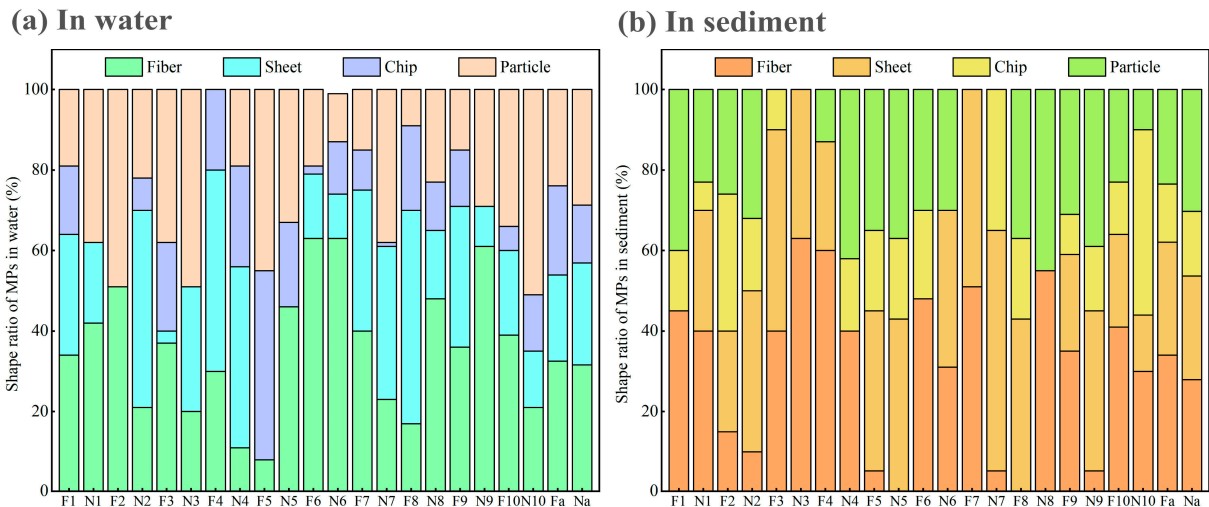

**Figure 3.** Shape characteristics of microplastics. "F1, F2 . . . F10" represent the proportion of MPs from the relevant sampling points (S1, S2 . . . S10) during the flood season; "N1, N2 . . . N10" represent the proportion of MPs from the relevant sampling points (S1, S2 . . . S10) during the non-flood season; "Fa" represents the average proportion of MPs across all sampling points during the flood season and "Na" represents the average proportion of MPs across all sampling points during the non-flood season. (**a**) in Water; (**b**) in sediment.

The proportion of fibrous MPs was the highest (27.90–34%) and the proportion of membrane MPs was the lowest (14.40–22.20%). The proportions of the MP shapes were relatively uniformly distributed, which is consistent with a study of the Wuhe River Basin in Poyang Lake and a study of the headwaters of Yangtze River [39]. Fibrous MPs can be attributed, to some extent, to sewage discharge from laundries [40]. Studies indicated that an average of more than 1900 fibers were produced by the cleaning of one cloth and more than 700,000 fibers were produced by a machine washing of 6 kg of acrylic clothes [41]. Li et al. reported that the removal rate of fibrous MPs by sewage treatment plants can reach 93.9%, but some fibers may still enter rivers after processing [42]. Fibrous plastics, which are also present in high proportions in air, can be precipitated directly with rainwater or enter into rivers via surface runoff, especially during flood seasons, dramatically increasing the concentration of fibrous MPs in a short period of time [43]. It is essential to note that the proportion of membrane MPs produced by agricultural planting or plastic packaging is not high. No anomalies in membrane MPs were detected at S8, which is located within an agricultural production area. This might indicate that recent measures to strengthen control over local agricultural plastic pollutants were relatively effective.

### 3.1.3. Size Characteristics of MPs

The size of the MPs, as scanned by LDIR, was divided into four categories: 0–500 µm, 500–1000 µm, 1000–2500 µm, and 2500–5000 µm. The size analysis of MPs is presented in Figure 4.

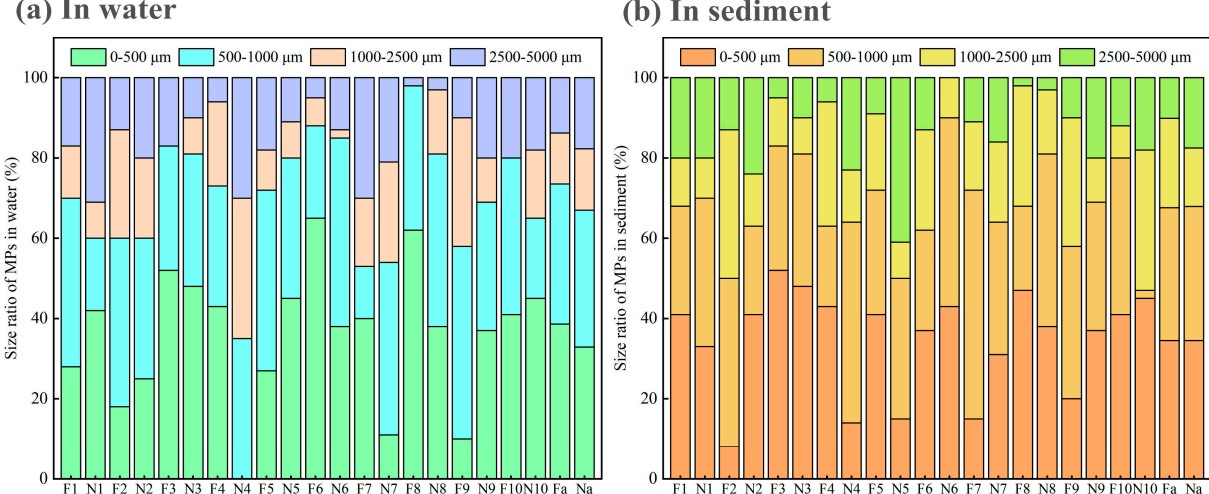

**Figure 4.** Size of microplastics determined by LDIR. "F1, F2 . . . F10" represent the proportion of MPs from the relevant sampling points (S1, S2 . . . S10) during the flood season; "N1, N2 . . . N10" represent the proportion of MPs from the relevant sampling points (S1, S2 . . . S10) during the non-flood season; "Fa" represents the average proportion of MPs across all sampling points during the flood season and "Na" represents the average proportion of MPs across all sampling points during the non-flood season. (**a**) in Water; (**b**) in sediment.

In water, the abundance of MPs sized 0–500 µm was the highest (38.60%) and the abundance of MPs sized 1000–2500 µm was the lowest (12.70%) during the flood season. During the non-flood season, the abundance of 500–1000 µm MPs was the highest (34.10%) and the abundance of 1000–2000 µm MPs was the lowest (15.30%). In sediments, the abundance of 0–500 µm MPs was the highest and the abundance of 2500–5000 µm MPs was the lowest (10.10%) during the flood season. During the non-flood season, the abundance of 0–500 µm MPs was the highest and the abundance of 2500–5000 µm MPs was the lowest (17.50%). There were equivalent proportions of MPs sized 0–500 µm and 500–1000 µm during the flood season and non-flood season. The statistical analysis revealed that the proportion of MPs sized 0–1000 µm was significantly higher (67–73.5%) when compared with those sized 1000–5000 µm, indicating that small-sized MPs were predominant in the Shiwuli River.

This is consistent with previous research results [44]. The volume of plastics decreases continuously in the natural environment due to secondary weathering, erosion, wearing, and degradation [45]. In the study by Murphy et al. [46], larger MPs were found to more easily coagulate and precipitate during sewage processing, resulting in a higher proportion of small-sized MPs being discharged into the river. Thus, the proportion of small-sized MPs that are finally discharged into rivers is relatively high. Further, MPs < 1 mm in size are easily making them attractive to aquatic organisms for ingestion and entering into the food chain, posing greater pollution threats [9,47]. There were more large MPs in the range of 1000–2000 µm and 2500–5000 µm during the non-flood season than during the flood season. It was reported that the size reduction of plastics in water bodies due to the hydraulic effect is negatively related to the stability of the water environment [48].

### 3.1.4. Colour Characteristics

The colours of MPs observed by microscope can generally be divided into seven types based on the colours white, red, green, blue, black, yellow, and hyaline in Figure 5.

**(a) In water**

**(b) In sediment**

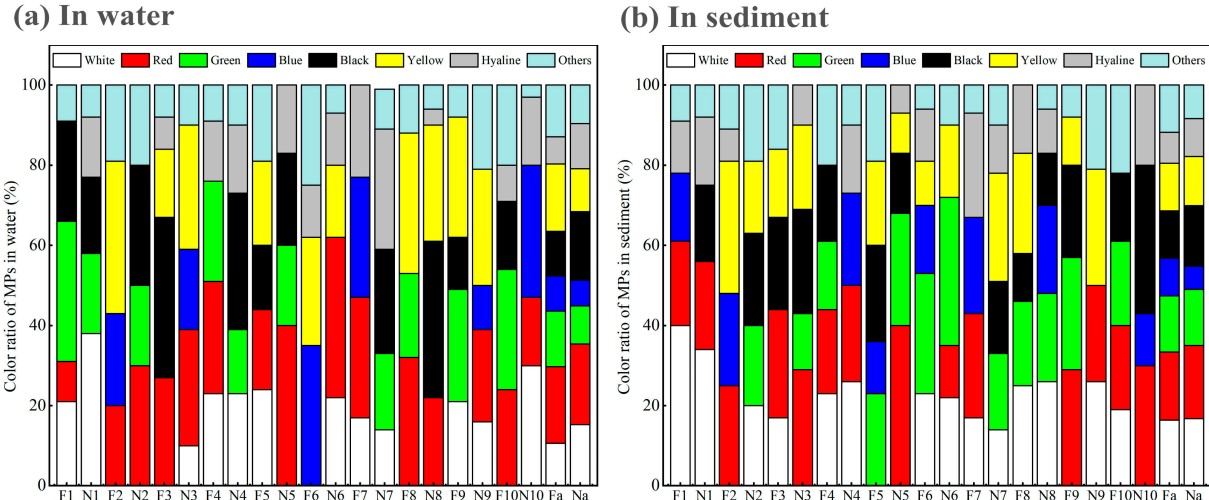

**Figure 5.** Colour characteristics of microplastics. "F1, F2 . . . F10" represent the proportion of MPs from the relevant sampling points (S1, S2 . . . S10) during the flood season; "N1, N2 . . . N10" represent the proportion of MPs from the relevant sampling points (S1, S2 . . . S10) during the non-flood season; "Fa" represents the average proportion of MPs across all sampling points during the flood season and "Na" represents the average proportion of MPs across all sampling points during the non-flood season. (**a**) in Water; (**b**) in sediment.

In water, the proportion of red MPs was the highest (19.10%) and the hyaline MPs was the lowest (6.80%) during the flood season. The proportion of red MPs also was the highest (20.10%) and the proportion of blue MPs was the lowest (6.40%) during the non-flood season. In sediments, the proportion of red MPs was the highest (17.00%) and the proportion of hyaline MPs was the lowest during the flood season. The proportion of red MPs was the highest (18.20%) and the proportion of blue MPs was the lowest (5.80%) during the non-flood season. The colours of the MPs indicated that the MPs were derived from extensive sources. The proportions of MPs of each colour were relatively uniform; however, the proportion of red MPs was the highest. The proportion of black MPs during the non-flood season was significantly higher than during the flood season. This might be because the MPs can absorb more pollutants and for longer in static water during the non-flood season, resulting in physical and chemical changes, which lead to discolouration [48,49]. The morphological observations revealed that the red MPs were mainly fibrous, the white and blue MPs were mainly particles, the green MPs were mainly fragments, and the yellow and hyaline MPs were mainly membranes.

Generally speaking, the colour distribution of MPs in the Shiwuli River Basin was relatively stable at the temporal scale, but there were differences in the spatial distributions. This situation is similar to the MP distribution in the Xiangxi River Basin [50]. The production and the consumption processes often involve colour modulation so as to improve attraction. Colourful MPs may fade over time and with external stress. Hence, the proportion of hyaline MPs in the sediment samples was significantly higher than in the water samples and the proportion of hyaline MPs during the non-flood season was much higher than during the flood season. In the fading process, common heavy metals in pigments might be released into the water, resulting in heavier pollution [51]. Moreover, it was demonstrated that small-sized colourful MPs are more easily attracted by aquatic animals and enter the human biological chain after being eaten, thus posing health risks [49,52].

### 3.1.5. Composition Characteristics

All particles in the samples from the Shiwuli River were scanned by LDIR, and after automatic comparison with the absorption peak profile of each MP polymer in the spectral library, a total of eleven MP polymers, such as acrylates (ACR), chlorinated polyethylene (CPE), ethylene vinyl acetate (EVA), polyethylene (PE), polyethylene terephthalate

(PET), polypropylene (PP), polystyrene (PS), polyurethane (PU), polyvinylchloride (PVC), polyamide (PA), and silicon, with matching degrees greater than 0.80, were identified. Six common types of polymer, PE, PP, PET, PA, PS, and PVC, as well as others, were selected for the proportional analysis. From Figure 6a, it can be seen that polyethylene (PE, 21.60%) dominates in water during the flood season, followed by polystyrene (PS, 15.60%). During the non-flood season, the proportion of PE (19.30%) was the highest, followed by the proportions of polypropylene (PP, 16.80%) and polyvinyl chloride (PVC, 15.50%). In sediments (Figure 6b), the proportion of PP (22.70%) was the highest during the flood season, followed by the proportions of PE (16.50%) and PVC (15.40%). The proportion of PE (23.20%) was the highest during the non-flood season, followed by PP (21.80%) and polyethylene terephthalate (PET, 18.10%). Generally speaking, the proportions of PE, PP, and PVC were relatively high, with the proportion of PE being the highest. This is consistent with other studies [34,53]. Examples of the qualitative results for the different types of polymer particles are shown in Figure 7.

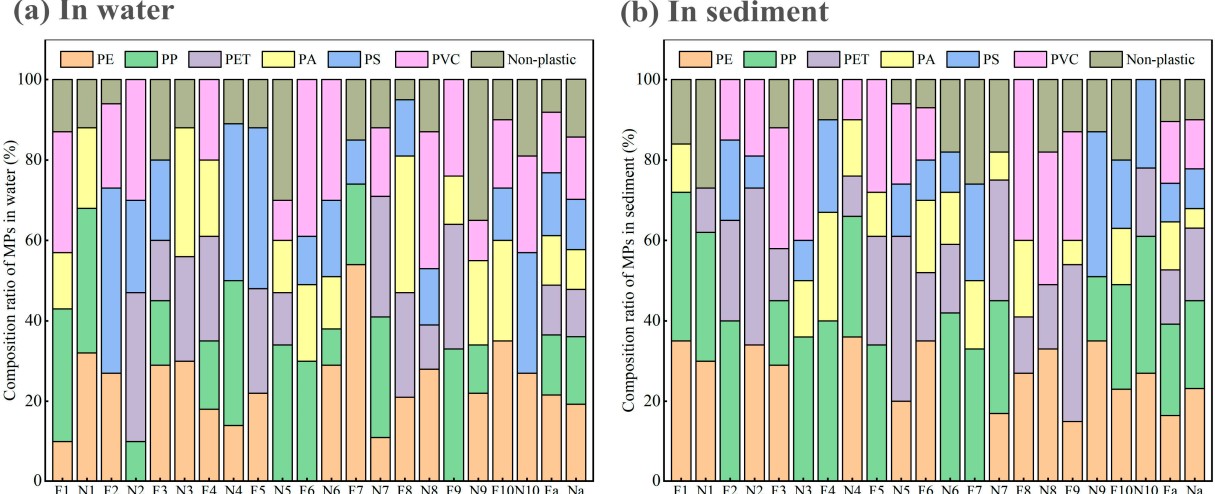

**Figure 6.** Composition characteristics of microplastics. "F1, F2 … F10" represent the proportion of MPs from the relevant sampling points (S1, S2 … S10) during the flood season; "N1, N2 … N10" represent the proportion of MPs from the relevant sampling points (S1, S2 … S10) during the non-flood season; "Fa" represents the average proportion of MPs across all sampling points during the flood season and "Na" represents the average proportion of MPs across all sampling points during the non-flood season. (**a**) in Water; (**b**) in sediment.

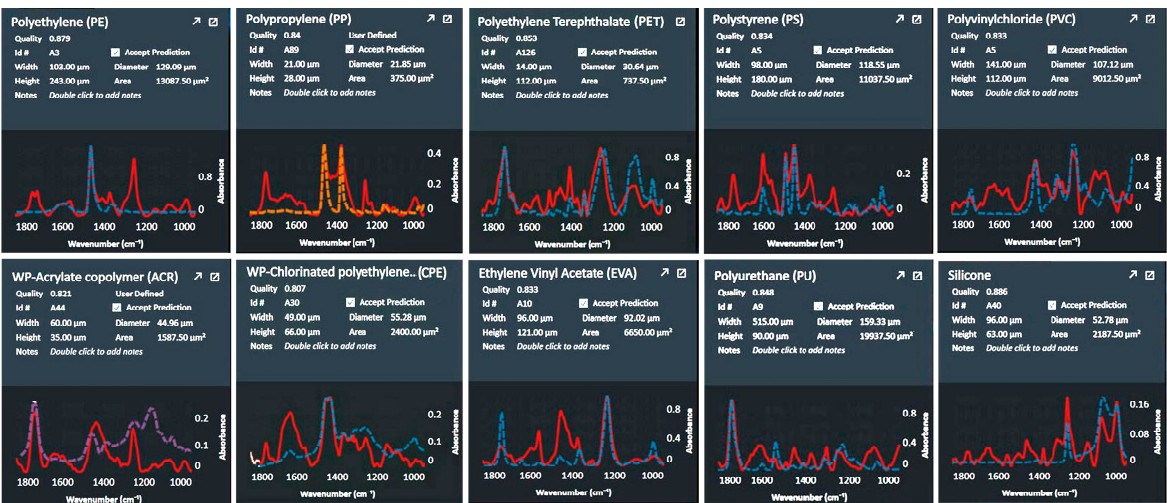

**Figure 7.** Curves of different particle types identified by LDIR (matching degree > 0.80).

Figure 8 shows the microstructure of common polymers scanned by SEM. The PEs were fragmented in shape (Figure 8a) with porous and rough surfaces. The PPs exhibited a membrane appearance (Figure 8b) with obvious fracture traces. The PETs were fragmented (Figure 8c) with rough cracks on the surface. The PAs were fibrous shapes (Figure 8d) with microfiber expenses on the surface. The PSs were particles (Figure 8e). The PVCs were fragments (Figure 8f) with obvious wearing. The detection rate in sediment samples was relatively high for PVC with a density of 1.38 $g/cm^3$;. The proportions of PP and PE, with low densities, were higher in the sediment samples, thus verifying their ability to adsorb and accumulate other pollutants [54]. This will lead to increases in the densities of PP and PE, resulting in their precipitation [55].

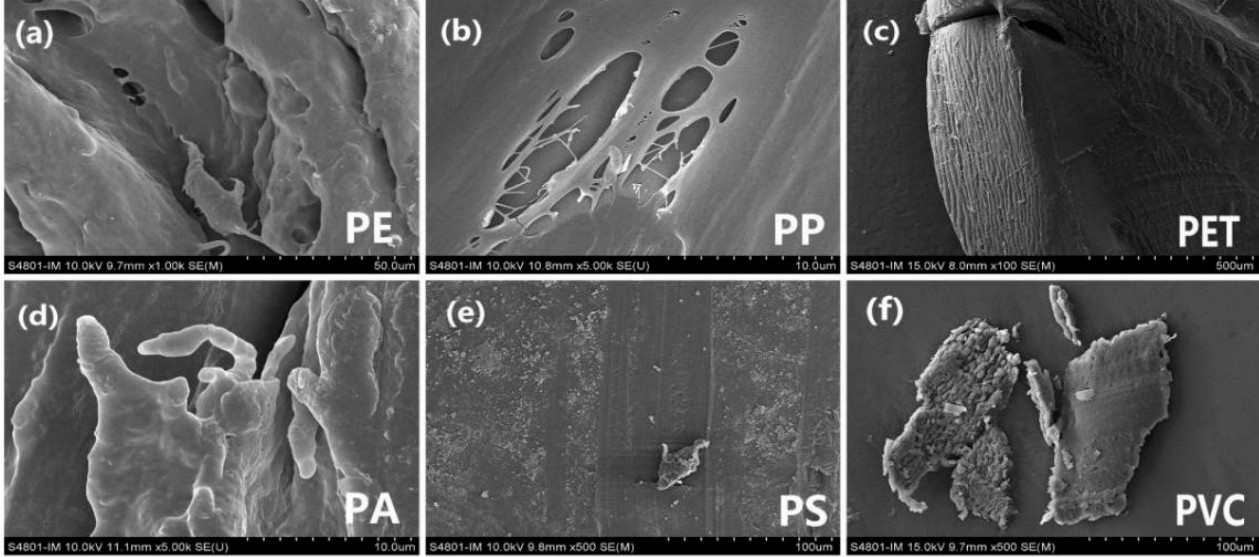

**Figure 8.** SEM images of common polymers. (**a**) PE; (**b**) PP; (**c**) PET; (**d**) PA; (**e**) PS; (**f**) PVC.

*3.2. Pollution Risk Assessment of Microplastics*

In the Shiwuli River, the risk index $H$ and the pollution load index $PLI_{zone}$ were calculated. The results are shown in Table 3.

**Table 3.** Risk assessment of microplastics in the Shiwuli River.

| | PE | | PP | | PET | | PA | | PS | | PVC | |
|---|---|---|---|---|---|---|---|---|---|---|---|---|
| $S_n$ [31] (hazard score, highest level) | 11 | | 1 | | 30 | | 50 | | 871 | | 30 | |
| Seasons (f.: flood; n.-f.: non-flood) | f. | n.-f. | f. | n.-f. | f. | n.-f. | f. | n.-f. | f. | n.-f. | f. | n.-f. |
| $P_n$ (%) | | | | | | | | | | | | |
| Water | 21.6 | 19.3 | 14.9 | 16.8 | 12.4 | 11.7 | 12.3 | 9.9 | 15.6 | 12.5 | 15.1 | 15.5 |
| Sediments | 16.5 | 23.2 | 22.7 | 21.8 | 13.5 | 18.1 | 11.9 | 4.8 | 9.6 | 9.9 | 15.4 | 12.2 |
| $H$ value | | | | | | | | | | | | |
| Water | 2.38 | 2.12 | 0.15 | 0.17 | 3.72 | 3.51 | 6.15 | 4.95 | 135.88 | 108.88 | 4.53 | 4.65 |
| Sediments | 1.82 | 2.55 | 0.23 | 0.22 | 4.05 | 5.43 | 5.95 | 2.40 | 83.62 | 86.23 | 4.62 | 3.66 |
| Risk level | | | | | | | | | | | | |
| Water | I | I | I | I | I | I | I | I | III | III | I | I |
| Sediments | I | I | I | I | I | I | I | I | II | II | I | I |
| $PLI_{zone}$ | | | | | | | | | | | | |
| Water | $PLI_{zone}$ (f.) value is 2.24 (heavy pollution); $PLI_{zone}$ (n.-f.) value is 1.66 (moderate pollution) | | | | | | | | | | | |
| Sediments | $PLI_{zone}$ (f.) value is 2.34 (heavy pollution); $PLI_{zone}$ (n.-f.) value is 1.91 (heavy pollution) [32] | | | | | | | | | | | |

The risks of PE, PP, PET, PA, and PVC in the water were classified as Level-I and the risk of PS was classified as Level-III. During the flood season, the $PLI_{zone}$ of MPs was

2.24 in water and 2.34 in sediments, both reflecting heavy pollution. During the non-flood season, the $PLI_{zone}$ of MPs was 1.66 in water and 1.91 in sediments, reflecting moderate pollution and heavy pollution, respectively. The risk index of PS indicated a higher risk degree in water than in sediments, whereas the $PLI_{zone}$ indicated that the pollution load in the sediments was higher. This is mainly attributed to differences in the reference indices between the two assessment models. The RI model uses the chemical toxicity of various polymer types as the major index, without consideration of the influences of the abundances of MPs. The PLI model uses the abundances of MPs as the major reference [31,32].

## 4. Conclusions

The spatial and temporal distribution characteristics of MPs in water and sediment samples from 10 representative points along the Shiwuli River in Hefei during both flood and non-flood seasons were studied. Additionally, an assessment of the ecological risks associated with MP pollution was conducted.

There was a difference in particle abundance among ten points. Results show that the water of the Shiwuli River had an average abundance of MPs of 8.4 ± 2.5 particles/L during the flood season and 5.8 ± 1.7 particles/L during the non-flood season; the sediment had an average abundance of MPs of 78.9 ± 8.3 particles/kg during the flood season and 63.9 ± 7.1 particles/kg during the non-flood season. The more abundance of MPs exhibited at confluences with tributaries (S4, S5, and S6) were close to the residential and industrial development, while lower values were identified in agricultural areas (S8) and wetland ecological regions (S9 and S10). In water, the maximum appeared at S5 with 21.7 ± 4.6 particles/L during the flood season and 15.9 ± 4.2 particles/L during the non-flood season, respectively; the minimum appeared at S9 with 1.8 ± 1.0 particles/L during the flood season and 2.2 ± 0.4 particles/L during the non-flood season, respectively. In sediment, the maximum appeared at S5 with 174.1 ± 10.1 particles/kg during the flood season and 143.6 ± 10.4 particles/kg during the non-flood season, respectively; the minimum appeared at S8 with 10.3 ± 2.8 particles/kg during the flood season and at S9 with 12.1 ± 3.2 particles/kg during the non-flood season, respectively. Analysis of MP characteristics showed that the MPs mainly exhibited a fibroid morphology (27.90–34%), likely originating from laundry activities and packaging breakage, and red-coloured particles (19.10%) within the smaller size less than 500 μm (38.60%) were more prevalent, making them attractive to aquatic organisms for ingestion. Additionally, results of MP identification show that a total of 11 types of polymers were found in the river water and sediment by using LDIR. According to matching analysis, these particles were identified as ACR, CPE, EVA, PE, PET, PP, PS, PU, PVC, PA, and silicone. Among them, PE emerged as the most prevalent polymer type (19.3−21.6%) due to its widespread use in daily life and relatively low cost.

The ecological risk assessment of different types of MP polymers revealed that during both flood and non-flood seasons, the risk levels of PS were classified as III in water and II in sediments in the Shiwuli River. However, for all other polymer types, the risk levels were categorized as I. The overall assessment of the MP pollution load of the Shiwuli River showed that during the flood season, the $PLI_{zone}$ value for water was 2.24, indicating heavy pollution, while during the non-flood season, it was 1.66, indicating moderate pollution. In the case of sediments, the $PLI_{zone}$ value was 2.34 during the flood season and 1.91 during the non-flood season, both indicating heavy pollution. These findings suggest that the MP pollution in the Shiwuli River poses a significant risk to the drinking water source of Chaohu Lake. This study aligns closely with other similar studies, as shown in Table 4.

**Table 4.** Comparison of the results of this study with other studies.

| Research Object | Country | Abundance | Assessment Models | Results of Assessment | References |
|---|---|---|---|---|---|
| Coast of India | India | 12.22–439 items/kg in sediment | Pollution load index (PLI) | *PLI* of west coast of India: 3.03–15.5 (heavy pollution) *PLI* of east coast of India: 1–6.14 (moderate to heavy pollution) | [21] |
| Chagan lake and Xianghai lake | China | Chagan Lake: $3.61 \pm 2.23$ particles/L; Xianghai lake: $0.29 \pm 0.11$ particles/L | Risk index (RI) | Levels-III (heavy pollution) in Chagan Lake and Xianghai Lake | [22] |
| Manas River Basin | China | $17 \pm 4$ items/L (April) $14 \pm 2$ items/L (July) | Risk index (RI) Pollution load index (PLI) | Most of the study areas: Level-III (heavy pollution) All the sampling sites: slightly polluted | [56] |
| Moheshkhali channel of Bangladesh | Bangladesh | Sediment: 138.33 items/m$^2$ Water: ~0.1 items/m$^3$ | Pollution load index (PLI) | $PLI_{sediments}$: 2.51 (heavy pollution) $PLI_{surface\ water}$: 1.67 (moderate pollution) | [57] |
| Shiwuli River (this study) | China | Water: Flood season (f.): $8.4 \pm 2.5$ particles/L Non-flood season (n.-f.): $5.8 \pm 1.7$ particles/L; Sediment: Flood season (f.): $78.9 \pm 8.3$ particles/kg Non-flood season (n.-f.): $63.9 \pm 7.1$ particles/kg. | Risk index (RI) Pollution load index (PLI) | PS: Level-III in water and Level-II in sediments; Other polymers: Level-I Water: $PLI_{zone}$ (f.): 2.24 (heavy pollution); $PLI_{zone}$ (n.-f.): 1.66 (moderate pollution) Sediments: $PLI_{zone}$ (f.): 2.34 (heavy pollution); $PLI_{zone}$ (n.-fl.): 1.91 (heavy pollution) | - |

The findings of this study indicate that further research into the degradation of MPs in several urban inland rivers that discharge into lakes that are the sources of drinking water is recommended. Emphasis should be placed on enhancing non-point source pollution control measures in urban river basins, including the implementation of infrastructure improvements, such as rainwater and sewage separation systems. These measures can effectively impede land-based MP entry into these rivers. Additionally, research should be conducted on effective engineering techniques for removing MPs, particularly those of smaller particle sizes, during sewage treatment processes. Urban inland rivers should be equipped with comprehensive water management systems to ensure ecological basic flow, and the natural degradation of MPs should be promoted through ecological methods, as is the case with constructed wetlands. From a regulatory standpoint, it is necessary to establish MPs risk assessment standards and conduct regular assessments of MP pollution in key rivers and lakes.

**Author Contributions:** Conceptualization, L.H. and T.B.; data curation, X.M. and Q.W.; investigation, X.M.; methodology, L.H. and B.L.; project administration, K.W.; resources, X.M. and Q.W.; supervision, K.W.; validation, L.H. and J.J.; writing—original draft, L.H.; writing—review and editing, T.B. All authors have read and agreed to the published version of the manuscript.

**Funding:** This research was funded by "Anhui innovation team for municipal solid waste treatment", "Anhui Key Laboratory of Sewage Purification and Eco-restoration Materials (YJ20230301-2)" and "National Special Item on Water Resource and Environment (2017ZX07603-003)".

**Data Availability Statement:** Not applicable.

**Conflicts of Interest:** The authors declare no conflict of interest.

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
