# Peer review of "Spatial–Temporal Distribution and Ecological Risk Assessment of Microplastics in the Shiwuli River"

_water, doi:10.3390/w15132330_

Round 1

Reviewer 1 Report

The paper “Spatial-temporal Distribution and Ecological Risk Assessment of Microplastics in the Shiwuli River “reports on an important topic and deserves publication. But before that some major issues need to be solved. In scientific research, it is crucial to articulate the purpose and relevance of the investigation being conducted. This serves to establish the significance and relevance of the study, and offers readers the necessary context to comprehend the rationale for the research. To communicate the importance of this study.

Q1. The introduction section of a manuscript should clearly elucidate the underlying information on the topic being researched and provide an explanation of the gap in knowledge that the study intends to bridge. This section should also specify the research objectives or questions that the study aims to answer and elucidate how the research will contribute to the advancement of the field.

Q2. Abstract section is confused and needs modification.

Q3.  In digestion process, is water samples were digested in autoclave or in open air? please clarify.

Q4.  Authors know the kind of microplastics in water by using Miro-FTIR, Where is the FTIR curves ?

Q5. Please measure TOC in all water samples.

Q6.Make relations between pH of water and type of microplastic present in water.

Q7. Make zonation map to show the distribution of microplastics in water.

Q8. Please compare the results between this works with other previous works .

Q9. The conclusion part is confused and needs modification.

Author Response

Dear Reviewer:
I sincerely thank you for your comments on my manuscript, they have been a guiding factor in improving this article and have increased my cognitive experience of the research.

For the response to your comments, please see the attachment.

Thank you again for your comments and I look forward to your continued support!

Reviewer 2 Report

The abstract needs to be edited. Some remarks are noted in the text. The paper could be published after minor changes.

The abstract needs to be edited. Some remarks are noted in the text. The paper could be published after minor changes.

Author Response

Dear Reviewer:

First of all, I sincerely thank you for your comments and suggestions on my manuscript, they have been a guiding factor in improving this article and have increased my cognitive experience of the research.

Your comments and suggestions were all focused on the abstract and introduction parts of the manuscript and I have therefore systematically modified the abstract, introduction and conclusion parts in response to your comments. Please understand that I have not responded to your comments point-by-point because of this.

Thank you again for your comments and I look forward to your continued support!

Reviewer 3 Report

The paper manuscript “Spatial-temporal Distribution and Ecological Risk Assessment of Microplastics in the Shiwuli River”

Overall, this is a well written manuscript that fit the journal scope section. Nevertheless, the authors should revise better their Results &  Discussion. and the  study must summarize and clearly present the findings  related to each method used and  how the findings relate to previous research in this area. 

In addition, several comments follow.

1) General Comment: Please check abbreviations with consistency in main text. Define it at the first appearance, then use it after the definition (e.g. MPs, PE, PLI, micro-FTIR,  etc.).

2) Risk levels are classified as Level I, II and III. Please provide further information including reference(s) regarding each classification. 

3) In addition regarding references [24-27], you could provide further more recent references.

4) Please confirm or update flood and non-flood given periods.

5) MPS separation and extraction: Additional data regarding the sampling methodology foe each type of matrix used in this study, is proposed to be reported. 

6) Concentration of MPs is given as n/L(Kg). Please express it as particle/L(Kg).

7) Regarding the quantification method please state if there are recovery data and provide them.  

8) It is proposed to provide representative image from microscope and FT-IR spectra for each polymer identified.

9) For figures 3,4 & 5 please clarify the symbols: F1,N1, F2,................N10, Fa, Na.

10) Describe the methodology used to determine colour characteristics. 

I will be glad to provide further details if needed and thank you for contacting me.

Author Response

Dear Reviewer:

First of all, I sincerely thank you for your comments and suggestions on my manuscript, they have been a guiding factor in improving this article and have increased my cognitive experience of the research.

For the response to your comments, please see the attachment.

Thank you again for your comments and I look forward to your continued support!

Round 2

Reviewer 1 Report

The manuscript titled" Spatial-temporal Distribution and Ecological Risk Assessment of Microplastics in the Shiwuli River"  can be accepted and suitable for publication in its present form.